

# The effects of warm air intrusions in the high arctic on cirrus clouds

Georgios Dekoutsidis[1], Martin Wirth[1], Silke Groß[1]

[1]Institut für Physik der Atmosphäre, Deutsches Zentrum für Luft- und Raumfahrt (DLR), Oberpfaffenhofen, 82234 Wessling, Germany

*Correspondence to*: Georgios Dekoutsidis (Georgios.dekoutsidis@dlr.de)

**Abstract.**

Warm Air Intrusions (WAI) are responsible for the transportation of warm and moist airmasses from the mid-latitudes into the high arctic (>70° N). In this work, we study cirrus clouds that form during WAI events (WAI cirrus) and during undisturbed arctic conditions (AC cirrus) and investigate possible differences between the two cloud types based on their macrophysical

and optical properties with a focus on Relative Humidity over ice (RHi). We use airborne measurements from the combined high spectral resolution and differential absorption lidar, WALES, performed during the HALO-(AC)3 campaign. We classify each research flight and the measured clouds as either AC or WAI, based on the ambient conditions and study the macrophysical, geometrical and optical characteristics for each cirrus group. As our main parameter we choose the Relative Humidity over ice (RHi) which we calculate RHi by combining the lidar water vapor measurements with model temperatures.

RHi is affected by and can be used as an indication of the nucleation process and the structure of cirrus clouds. We find that during WAI events the arctic is warmer and moister and WAI cirrus are both geometrically and optically thicker compared to AC cirrus. WAI cirrus and the layer directly surrounding them, are more frequently supersaturated, also at high supersaturations over the threshold for homogeneous ice nucleation (HOM). AC cirrus have a supersaturation dominated cloud top and a subsaturated cloud base. WAI cirrus also have high supersaturations at cloud top, but also at cloud base.

## 1 Introduction

Global warming, the increase of the average global temperature is a well-studied and proven fact (e.g. Hansen et al., 2010; IPCC, 2021). In recent decades scientists have noticed that the Arctic regions are warming much faster than the global average, a phenomenon named Arctic Amplification (Serreze and Francis, 2006; Graversen and Wang, 2009; Hansen et al., 2010; Wendisch et al., 2023). The exact causes, processes and contributions of different atmospheric parameters to this phenomenon

are a matter of ongoing research (e.g. Winton, 2006; Graversen and Wang, 2009; Pithan et al., 2018; Stuecker et al., 2018; Wendisch et al., 2019, 2023). Some studies are identifying increased poleward atmospheric transport into the arctic as one of the reasons (Park et al., 2015; Binder et al., 2017; Dahlke and Maturilli, 2017).

Such airmass transports are called Warm Air Intrusions (WAI). During WAI events warm, water-vapor- and aerosol-rich airmasses are meridionally transported from the mid-latitudes into the otherwise cold, dry and 'clean' arctic (Doyle et al.,

2011; Stramler et al., 2011; Woods and Caballero, 2016; Liu et al., 2018; Pithan et al., 2018; Hartmann et al., 2019, 2021; Beer



et al., 2022; Dada et al., 2022). Apart from the apparent transfer of sensible heat and the resulting warming, these events also affect the radiative balance in the arctic, which can lead to further warming. They increase the amount of water vapor, an important greenhouse gas, the amount of cloud condensation nuclei (CCN) and to a lesser degree the amount of ice nucleating particles (IN) (Dada et al., 2022), since only a very small amount of aerosols are suitable as IN (DeMott et al., 2003; Pruppacher

and Klett, 2010) . This leads to an enhancement of the cloudiness and affects the longevity of clouds (Dada et al., 2022; Kapsch et al., 2013; Loewe et al., 2017; Mitchell et al., 2018) and in turn affects the radiation balance as for example cirrus clouds that form can have a warming effect by increasing the downward longwave radiation (Shupe and Intrieri, 2004; Doyle et al., 2011; Stramler et al., 2011; Park et al., 2015; Mitchell et al., 2018; Pithan et al., 2018; Yamanouchi, 2019; Dada et al., 2022). Studying these events is becoming increasingly more important as studies have shown that there is an increase in the amount and duration

of WAI events in the arctic, with a projected continuing positive trend in the future, leading to them playing an ever more important role in the arctic climate (Woods and Caballero, 2016; Graham et al., 2017; Henderson et al., 2021).

Cirrus clouds play an important role in the regulation of water vapor distribution and uptake for droplet and ice crystal formation, as well as in the radiative processes of the upper troposphere and lower stratosphere (UTLS), with a significant effect on the climate (Liou, 1986; Krämer et al., 2016; Dekoutsidis et al., 2023). During nighttime all clouds induce a positive

cloud radiative forcing (CRF) at top of the atmosphere (TOA), causing warming. During daytime, because of their unique nature, cirrus is the only cloud type that can induce a positive CRF at TOA, depending on internal microphysical and external parameters such as solar zenith angle, temperature, altitude and others (Gasparini et al., 2018; Campbell et al., 2021; Marsing et al., 2023). The exact radiative effects of cirrus clouds are still not well quantified, although many studies conclude that they present an all-year net warming globally and at high latitudes (Hong and Liu, 2015; Gasparini and Lohmann, 2016; Hong et

al., 2016).

All cirrus clouds form via homogeneous (HOM) or heterogeneous (HET) ice nucleation, or rather a combination of the two processes with one being dominant. The nucleation process of each cloud is largely dictated by the ambient conditions, e.g. vertical speed omega, available INPs, available moisture and ambient temperature. Depending on the nucleation process, the microphysical characteristics of the clouds are different, which in turn leads to differences in the macrophysical and radiative

properties (Comstock et al., 2008; Mitchell et al., 2018, 2020; De La Torre Castro et al., 2023). Based on this, we expect cirrus that form during WAI events to have different formation processes, characteristics and effects compared to cirrus that form in undisturbed arctic conditions. Studying the geometrical and optical properties of cirrus under varying conditions is a prerequisite to understanding their overall effect on the climate and arctic amplification.

Despite the abovementioned, the arctic is a region where atmospheric measurements are sparser than at lower latitudes, be it

water vapor, aerosols, INPs, clouds or supersaturation. Direct measurements of WAI events in the arctic and especially of cirrus clouds during these events are also lacking (Wendisch et al., 2019). This hinders the study of one of the components expected to have a major impact on arctic amplification. Additionally, even in existing studies about the changing arctic, cirrus clouds are frequently overlooked (Ali and Pithan, 2020; Bossioli et al., 2021; Graham et al., 2017; Park et al., 2015). Due to





this lack of data, models have been used, but still many of them use simple parametrizations, which also need measurements in order to be verified and further developed (Wendisch et al., 2019).

The HALO-(AC)3 campaign was conducted in spring of 2022 aiming to fill this observational gap (Ehrlich et al., 2023; HALO AC3, 2023). In this work we use airborne lidar measurements of cirrus clouds conducted in the arctic regions during this campaign. Airborne lidar measurements have been shown to be suitable for the study of cirrus clouds (Dekoutsidis et al., 2023; Groß et al., 2014). During the campaign the flight planning was conducted in such a way that airmasses and clouds were measured both under undisturbed arctic (AC) conditions as well as during WAI events. We group the measured clouds accordingly into arctic (AC) cirrus and warm air intrusion (WAI) cirrus depending on the ambient conditions. Our aim is to study how the arctic region and the cirrus clouds that form there are affected by WAI events with respect to their geometrical, optical and macrophysical properties.

We are specifically interested in the relative humidity over ice (RHi). Ice supersaturation, RHi>100%, commonly occurs inside and around cirrus clouds, especially in the arctic regions (Ovarlez et al., 2002; Spichtinger and Gierens, 2009; Krämer et al., 2016; Gierens et al., 2020; Dekoutsidis et al., 2023). It is a prerequisite for the formation of any ice cloud either by HOM or HET nucleation. Nevertheless, the distribution of RHi in the resulting cloud is highly dependent on the nucleation process, as for example HOM needs a much higher RHi to initiate (Koop et al., 2000; Kärcher and Lohmann, 2002, 2003; Haag et al., 2003; Krämer et al., 2009; Spichtinger and Gierens, 2009; Kärcher, 2012; Kärcher et al., 2022). Thus, by studying the RHi inside and in the vicinity of cirrus clouds we can also gain some insights in the nucleation processes (Kärcher and Lohmann, 2002, 2003; Haag et al., 2003; Urbanek et al., 2017; Dekoutsidis et al., 2023). RHi is also important on the evolution and life cycle of cirrus, as it is guiding processes such as deposition and sublimation of ice crystals and regulates water vapor uptake for ice crystal nucleation (Kärcher, 2012; Zhao and Shi, 2023). Despite its importance, the exact mechanisms controlling the RHi are not well understood and although ice nucleation parametrizations that take supersaturation into account have been developed, some global models still do not allow high ice supersaturations (Kärcher and Lohmann, 2002, 2003; Haag et al., 2003; Lohmann et al., 2004; Liu et al., 2007; Comstock et al., 2008).

## 2 Data and method

### 2.1 HALO research aircraft and HALO-(AC)3 campaign

The German High Altitude and LOng-range research aircraft (HALO) is a platform for airborne measurements of the atmosphere. With a maximum range of 12500 km, a flight ceiling of 15.5 km and a maximum payload of 3 tons it is capable of transporting the necessary instrumentation and personnel to remote polar regions while also reaching the high altitudes, making it an ideal platform for the study of cirrus clouds in the arctic (Krautstrunk and Giez, 2012; Groß et al., 2014).

HALO was employed for the HALO-(AC)3 field campaign during March/April 2022 (Ehrlich et al., 2023; HALO AC3, 2023). Aim of the campaign was among others to investigate Warm Air Intrusions (WAI) into the arctic. HALO was equipped with state-of-the-art, active and passive remote sensing instrumentation and was stationed in Kiruna, Sweden (Ehrlich et al., 2023).



A total of 18 research flights were performed covering a vast area in the arctic between North Sweden, the North Pole and Greenland during which airmasses and clouds with various origins and characteristics were measured. In this study we use data retrieved during 15 of these research flights, excluding the transfer flights as well as flights with ambiguous classification or too few measured clouds. In Fig. 1 we present the water vapor mixing ratio measured along the flight tracks of these flights,
grouped into two groups: Left: flights during which undisturbed arctic conditions where prevailing and Right: flights during prevailing WAI events.

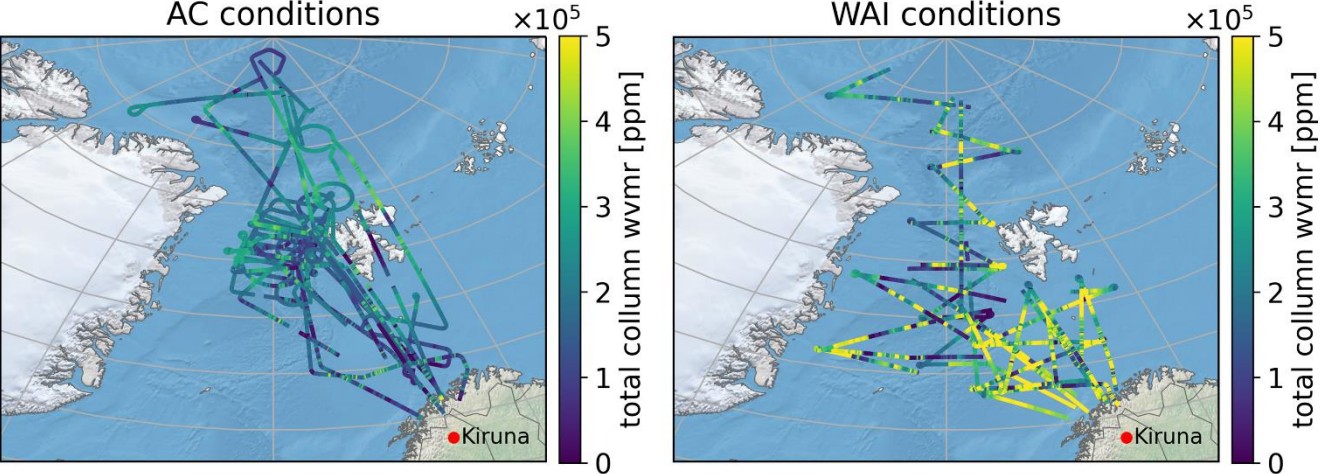

**Figure 1: Total column water vapor mixing ratio measured along the flight tracks of 15 flights during the HALO-(AC)3 campaign in March/April 2022. Left: Flights during undisturbed arctic conditions (AC). Right: Flights during an active Warm Air Intrusion (WAI) event. Background map tiles, borders and coastlines are by Natural Earth (naturalearthdata.com). License: Public domain**

### 2.2 WALES lidar system

The WAter vapor Lidar Experiment in Space (WALES) lidar system is a state-of-the-art instrument developed and operated by the Institute of Atmospheric Physics (IPA) at DLR (Wirth et al., 2009). It is a combined Differential Absorption Lidar
(DIAL) and High Spectral Resolution Lidar (HSRL) system (Ehret et al., 1993; Esselborn et al., 2008). It can provide collocated measurements of the extinction coefficient, backscatter coefficient, linear particle depolarization ratio and water vapor mixing ratio. For the DIAL technique laser pulses are emitted at four wavelengths in the 935 nm absorption band of water vapor, one off-line and three on-line. The depolarization is measured by a cross-polarized channel at 532 nm wavelength. The 532 nm signal is also used for the extinction coefficient measurements by the HSRL method (Ehret et al., 1993; Esselborn
et al., 2008).

In this study we use the measured backscatter coefficient and depolarization ratio for the construction of our cloud mask, the optical transmission to calculate the optical depth and the water vapor mixing ratio in the calculation of the RHi. An advantage of airborne lidar measurements in general is that the measurements come in form of a two-dimensional curtain covering almost the whole atmospheric column from the aircraft to the earth's surface when conditions are favorable. Thus, with a single
overpass of a cloud information about its vertical and horizontal structure is provided. The total flight time in AC and WAI



conditions and the total number of HSRL and DIAL datapoints is given in Table 1. Another advantage of WALES is the accuracy of the measurements. Kiemle et al. (2008) estimate the statistical error in the retrieval of the water vapor to be around 5%. The exact value for each measurement might differ as it depends on various parameters but is estimated and tabulated in the datasets. During the HALO TECHNO mission, WALES performed water vapor measurements from HALO, while a second

aircraft was conducting simultaneous in-situ measurements at a lower altitude. Groß et al. (2014) compared the WALES DIAL to the collocated in-situ measurements (their Fig. 4) and found a deviation of < 1 % between the two data sets when the horizontal separation of the two aircraft was small. In the same publication they also highlight the suitability of WALES for the study of cirrus due to its capability of resolving even small-scale features of the clouds (Groß et al., 2014).

**Table 1: Amounts of WALES data used in this study. Data points grouped according to the prevailing synoptic situation.**

|  | Arctic (AC) conditions | Warm Air Intrusions (WAI) |
|---|---|---|
| Total time of flights (HH:MM:SS) | 61:08:24 | 43:40:12 |
| Total amount of data HSRL (in-cloud) | 9'652'871 (808'073) | 6'448'923 (1'381'828) |
| Total amount of data DIAL (in-cloud) | 8'418'723 (687'481) | 4'330'523 (942'111) |

**2.3 Methodology**

**2.3.1 Flight Grouping**

In this study we aim to detect differences in cirrus clouds measured in the high arctic during undisturbed arctic conditions (AC) and during prevailing warm air intrusion conditions (WAI). We studied the synoptic situation for each flight, using the geopotential height, temperature and total column water vapor fields from the ERA5 reanalysis dataset (Hersbach et al., 2018).

Based on this analysis each flight was characterized as being under AC or WAI conditions and grouped accordingly (Fig. 2). Henceforth, we shall refer to the cirrus clouds of the AC and WAI groups as arctic cirrus and WAI cirrus respectively. In order to be more precise in our classification parts of some flights where discarded, mostly due to HALO exiting the WAI events when performing turns.

**2.3.2 Ambient conditions**

After performing this initial analysis and creating the two groups, we used again the temperature and water vapor fields from ERA5 as well as the water vapor mixing ratio measurements by WALES to quantify the differences between the two groups. For the ERA5 fields we calculated the means in a box with coordinates -30 E to 30 E and 70 N to 85 N which contains the region where the field campaign took place. For the temperature field we used the data on the pressure levels at: 250, 500, 850 and 1000 hPa in order to cover the whole atmospheric profile. The means of each atmospheric column for the water vapor

mixing ratio measured by WALES were calculated along the flight track of each flight.



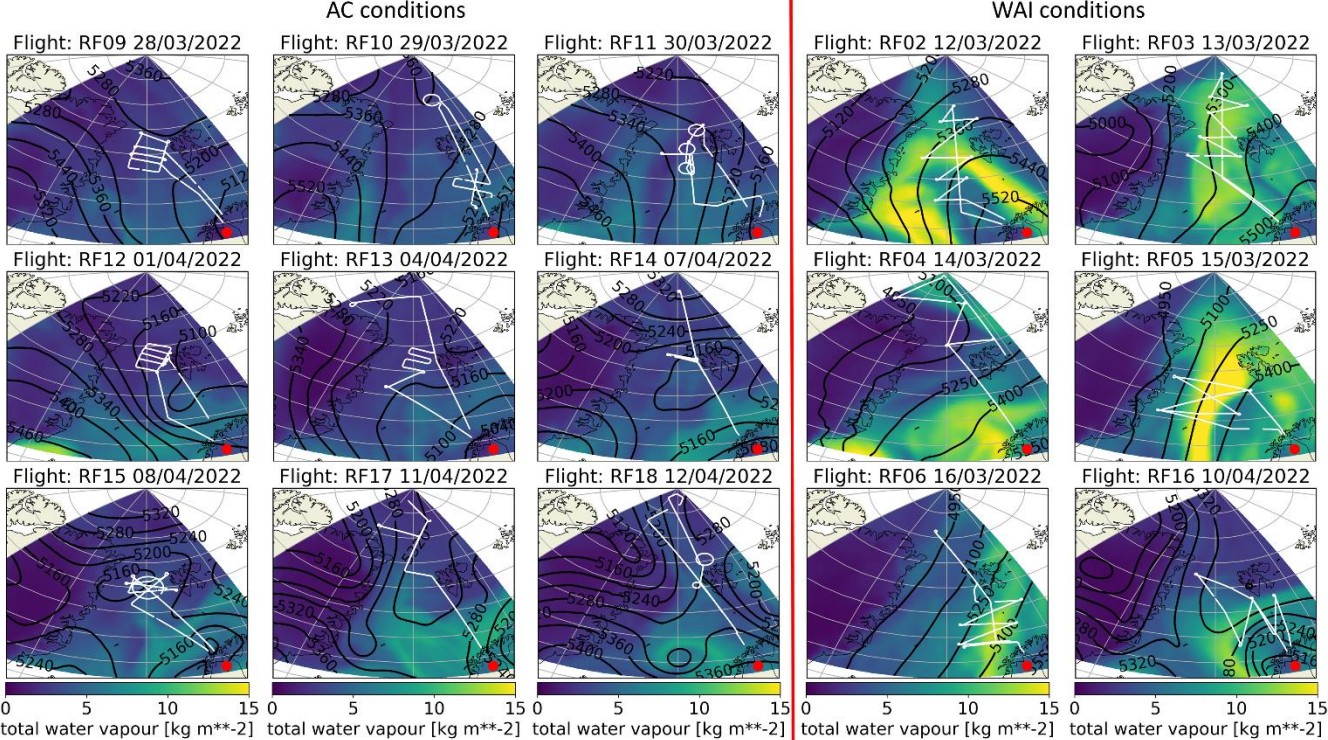

**Figure 2: In color we present the total column water vapor and the contours represent the geopotential height at 500 hPa in gpm at temporal midpoint for the HALO-(AC)3 research flights used in this study. Data from ERA-5. Also depicted the flight track for each flight. The flights to the left of the red line are the arctic condition (AC) cases and to the right of the red line the WAI cases.**

### 2.3.3 Relative Humidity over ice (RHi)

In order to calculate the RHi we use the water vapor measurements from WALES and model temperatures from the European Centre for Medium Range Weather Forecasts (ECMWF). To obtain temperature data at the measurement location ECMWF
operational analysis and short-term forecast fields from the atmospheric high-resolution model (HRES, 0.125x0.125 grid, 137 model levels) are spatially and temporally interpolated to the observation locations. For details about the data and interpolation the interested reader is referred to Schäfler et al. (2020). We apply the formula from Huang (2018) to calculate the saturation vapor pressure over ice from which we then find the RHi. This method has also been used by Groß et al., (2014), Urbanek et al., (2017, 2018) and Dekoutsidis et al., (2023). Groß et al. (2014), additionally perform an in-depth analysis on the suitability
of the ECMWF model temperatures and the resulting errors. They compare the model temperatures with in-situ aircraft measurements and find a height independent difference of 0.8K ($T_{HALO} - T_{ECMWF}$) in the usual altitude range of cirrus clouds. They conclude that this leads to an uncertainty of 10-15% in the final calculated RHi, but consider ECMWF model temperatures suitable for this type of study. Wirth and Groß, 2023 compared the ECMWF temperatures with dropsonde data



from the HALO-(AC)3 campaign and also come to the same conclusion. Since we use data from the same instrument and a similar methodology we consider that the same applies also for our study.

### 2.3.4 Masking

During each flight various clouds and cloud types are measured as well as the cloud-free atmosphere. In order to keep only the cirrus clouds, we created and applied a cloud mask to our data (Table 2). We defined three regions: In-cloud, near-cloud and cloud-free. For the definition of the in-cloud data points we used the backscatter ratio (BSR), particle linear depolarization (PLDR) and altitude (H). For the BSR we performed an analysis and found a BSR threshold of 2 to be fitting for this study. Groß et al. (2014), Urbanek et al. (2017 and 2018) and Dekoutsidis et al. (2023) found a wide range of BSR values between 2 and 25 to be suitable to define the in-cloud data, without significantly affecting the results. For the PLDR we chose a threshold of 20% as this is representative for ice crystals (Gobbi, 1998; Chen et al., 2002; Comstock et al., 2004). Finally, we chose to keep only data above an altitude of 3000 m in order to remove any lower clouds. The near-cloud data points form a band around the previously defined cloud borders. This band reaches 80 km before and after the cloud edges and 500 m above/below the cloud top/base. The cloud-free area consists of all the data beyond the near-cloud area. An example of the data and the resulting three areas can be seen in Fig. 3.

**Table 2: Thresholds for the in-cloud, near-cloud and cloud-free masks**

| | |
|---|---|
| In-cloud | BSR ≥ 2 |
| | PLDR ≥ 20 % |
| | H ≥ 3000 m |
| Near-cloud | 80 km before and after cloud edges |
| | 500 m over/under cloud top/bottom |
| | H ≥ 3000 m |
| Cloud-free | remaining data points |
| | H ≥ 3000 m |

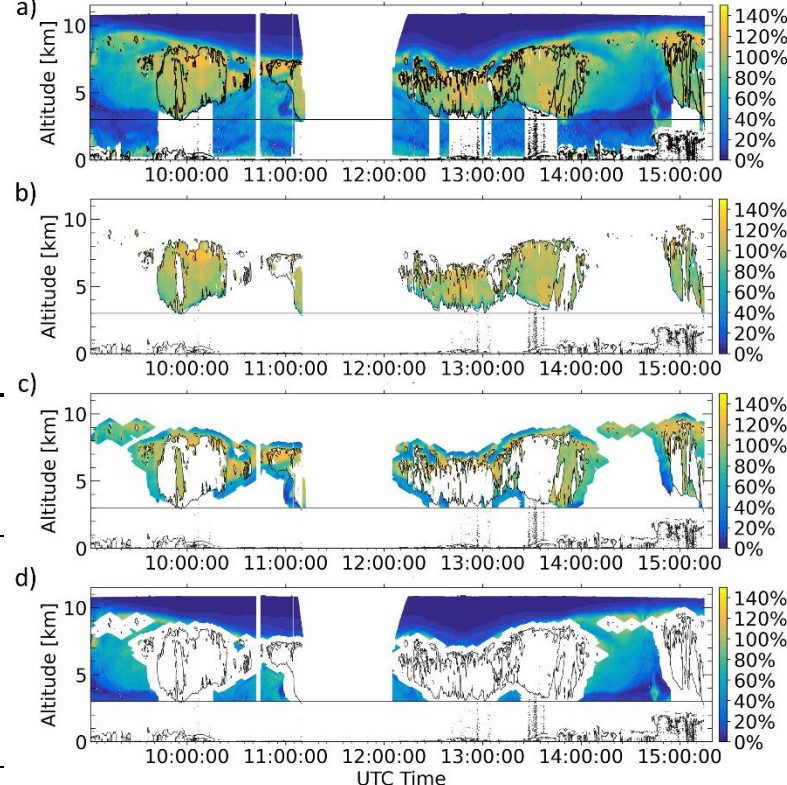

**Figure 3: Example of cloud mask application. (a) Complete RHi curtain with cloud mask outline (b) In-cloud data (c) Near-cloud data and (d) cloud-free data. The black lines correspond to the mask parameters. The color scale represents RHi. White areas have no data. HALO-(AC)3 campaign research flight 17 on 11/04/2022**





### 2.3.5 Nucleation processes

The nucleation process of cirrus can be either homogeneous (HOM) or heterogeneous (HET). HOM nucleation occurs when supercooled solution droplets (SSPs) freeze at low temperatures (< 235 K), but very high supersaturations are necessary (Koop et al., 2000; DeMott et al., 2003; Pruppacher and Klett, 2010). For HET nucleation to take place, ice-nucleating particles (INPs) are needed (DeMott et al., 2003; Pruppacher and Klett, 2010). INPs with different characteristics have different freezing thresholds. Easily activated INPs can initiate the formation of ice crystals already at low supersaturations (DeMott et al., 2003; Kärcher and Lohmann, 2003). Here, we calculate three temperature-dependent thresholds: one for HOM nucleation and two for HET nucleation. (Urbanek et al., 2017, their Table 1, and original formulations from Krämer et al., 2016). For the higher HET threshold, we consider coated soot (CS) as the INP, which is ineffective and thus makes higher RHi values necessary. For the lower HET threshold, we choose mineral dust (MD), which is more efficient as an INP needing lower RHi (DeMott et al., 2003; Kärcher and Lohmann, 2002). Further, we define three RHi regimes. The first one for RHi values over 147 %, in which HOM nucleation would be possible (HOM regime) and two regimes for RHi values where HET nucleation would be possible (high and low HET regime). For ease of understanding and in order to generalize beyond MD and CS we define the low HET regime from 100 % to 123 % RHi for easily activated INP and the high HET regime from 123% to 147 % RHi for not so effective INP, effectively splitting the HET range in half.

### 3 Results

### 3.1 The high arctic under AC and WAI conditions

The general state of the atmosphere in the high arctic is bound to change significantly during a WAI event as warm and moist airmasses are introduced into the otherwise cold and dry region. In Table 3 we present a quantification of these differences from our analysis. We start by looking into the total column water vapor and temperature fields from the ERA-5 reanalysis dataset. As expected, there is an increase in both the mean amount of water vapor present in the high arctic and the temperature during the days with a prevailing WAI event. On average the available water vapor doubles from 3.4 to 6.9 kg·m$^{-2}$ between AC and WAI conditions respectively. Regarding the temperature we present an analysis on four pressure levels, from near-surface (1000 hPa) up to the 250hPa pressure level. We find an average surface warming of almost 10 K during WAI events. Higher temperatures are detected on average during WAI events up to the 500hPa pressure level, with the differences becoming progressively smaller for higher altitudes until at the 250 hPa pressure level the arctic becomes on average slightly warmer on AC condition days, in contrast to the lower altitudes. A probable explanation for this observation is, that due to the colder AC conditions the tropopause is lower and the 250 hPa pressure level is already in the stratosphere, where the temperature tends to increase with altitude.

In Table 1 we show that WALES collected a significant amount of data during the research flights performed under both AC and WAI conditions, in total as well as in-cloud. This gives us the confidence to perform statistical analyses and extract results




representative of cirrus clouds in the spring high arctic. Thus, after studying the ERA5 data we further augment our analysis by using the measurements taken by WALES during the HALO-(AC)3 campaign (Table 3). Similar to the findings from ERA5, the mean water vapor mixing ratio measured during research flights with prevailing WAI events is found to be more than doubled compared to flights under AC conditions, with 888 and 352 ppm respectively. Since WALES provides a two-dimensional dataset we are further analyzing the water vapor mixing ratio on four vertical levels, from earth's surface up to an altitude of 12 km. We find that throughout the atmospheric profile there is around 2.5 times more water vapor in the arctic troposphere during prevailing WAI events compared to flights under AC conditions.

## 3.2 Cirrus clouds under AC and WAI conditions

### 3.2.1 Geometrical and optical characteristics

The geometrical characteristics of cirrus clouds measured during flights under AC and WAI conditions also differ to each other as we show in Table 3. Cirrus clouds are on average geometrically thicker when measured during a WAI event than during AC conditions with a mean geometric depth of 3141 m and 2037 m respectively. When considering the cloud base heights, we find that the cloud base of WAI cirrus is higher than that of AC cirrus by an average of around 652 m (4990 m and 4338 m respectively). Similarly, the cloud tops of WAI clouds reach much higher than the tops of AC cirrus by an average of around 1755 m with 8131 m and 6376 m respectively. Hence, the cloud base heights of arctic and WAI clouds are closer to each other than the average cloud top heights. It must be kept in mind that in some cases the cloud bases might have been excluded due to the altitude limit of the cloud mask. Additionally, in some cases the lidar signal might become completely attenuated before reaching the actual cloud base. In that case the cloud base is considered the lowest altitude where a measurement was taken. The cloud-base heights defined by the lidar measurements were compared to the ones measured by the HALO Microwave Package (HAMP) cloud radar. An average difference (WALES -HAMP) of 388 m for WAI clouds and -118 m for AC cirrus was found. The bigger vertical extent of the WAI cirrus can also be attributed to the generally expected higher tropopause altitude during WAI events due to the warmer ambient temperatures.

Finally, we have also calculated the average optical depths for AC and WAI cirrus. As would be expected also due to the differences in their geometrical extent, WAI cirrus have an almost two times greater optical depth than AC cirrus with 2.48 and 1.39 respectively.

### 3.2.2 Distribution of RHi

In Fig. 4 we present the 2-D probability densities of RHi with respect to ambient temperature for the in-cloud, near-cloud and cloud-free regions of the research flights in the AC and WAI groups (See Sect. 2.3.3 and. Sect 2.3.1). Additionally, for each panel we plot the ice saturation threshold (RHi = 100%), and three temperature dependent nucleation thresholds. Two for HET nucleation, one for mineral dust (MD) as INP and one for coated soot (CS) as well as the threshold for HOM nucleation (See Sect. 2.3.4).





Starting with the in-cloud data for the two groups, panels (a) and (d), we find that the peak of the probability density for the in-cloud data of the AC cirrus group is at an RHi of 100 % and a temperature of 242 K. For the WAI cirrus the peak probability

is at a similar RHi of 99% and a slightly higher temperature of 246 K. Thus, the highest frequency of detection for both cloud types is around the saturation level. WAI cirrus are detected in a wider range of temperatures than AC cirrus both on the warmer and the colder end. WAI cirrus also appear to be more frequently supersaturated than arctic cirrus. Both cloud types have a significant amount of data points over the activation threshold of mineral dust. While WAI cirrus also reach up to the activation threshold for coated soot almost throughout the whole temperature range, the AC cirrus only reach that threshold

on a narrower temperature range. Inside WAI cirrus the frequency of detection over the activation threshold for CS is significantly higher than for the AC cirrus. The same applies also for the HOM threshold, with the WAI cirrus having a lot more data points over it compared to AC cirrus.

**Table 3: Differences in available water vapor, temperature and cloud geometric and optical characteristics during research flights of the HALO-(AC)3 campaign under arctic (AC) and warm air intrusion (WAI) conditions. Overview from ERA-5 reanalysis data**
**and data from WALES measurements.**

|  | AC | WAI | WAI-AC | Difference % |
|---|---|---|---|---|
| ERA-5 means |  |  |  |  |
| Total column water vapor (kg·m$^{-2}$) | $3.4 \pm 0.4$ | $6.9 \pm 1.9$ | 3.5 | 103.3 |
| Temperature (K) at: |  |  |  |  |
| 250 hPa | $219 \pm 5$ | $216 \pm 6$ | -3 | -1.6 |
| 500 hPa | $237 \pm 2$ | $242 \pm 4$ | 5 | 2.2 |
| 850 hPa | $257 \pm 1$ | $266 \pm 3$ | 8 | 3.2 |
| 1000 hPa | $260 \pm 1$ | $270 \pm 4$ | 10 | 3.8 |
| WALES means |  |  |  |  |
| Total water vapor mixing ratio (ppm) | $352 \pm 93$ | $888 \pm 280$ | 536 | 152.4 |
| Water vapor mixing ratio (ppm) for: |  |  |  |  |
| 9-12 km | $8 \pm 2$ | $22 \pm 10$ | 14 | 171.0 |
| 6-9 km | $70 \pm 32$ | $177 \pm 49$ | 107 | 152.0 |
| 3-6 km | $382 \pm 107$ | $908 \pm 183$ | 526 | 137.4 |
| 0-3 km | $947 \pm 149$ | $2446 \pm 528$ | 1499 | 158.3 |
| Cloud characteristics |  |  |  |  |
| Cloud geometrical depth (m) | $2037 \pm 855$ | $3141 \pm 734$ | 1104 | 54.2 |
| Cloud top height (m) | $6376 \pm 1195$ | $8131 \pm 1229$ | 1755 | 27.5 |
| Cloud base height (m) | $4338 \pm 609$ | $4990 \pm 1284$ | 652 | 15.0 |
| Optical depth | $1.39 \pm 0.34$ | $2.48 \pm 0.98$ | 1.09 | 78.4 |




Regarding the airmasses immediately bordering the clouds of the two groups (Fig. 4 panels (b) and (e)), the temperature ranges are similar to the in-cloud data points of each group. Judging by the contours we see that there is still a high frequency of supersaturated data points, but most are subsaturated with significant amounts reaching values close to 0 % RHi, especially

for the WAI clouds. These values might be a result of the near-cloud area extending into the stratosphere or an indication of complete depletion of the available water vapor during cloud formation. Regarding the supersaturated data points, we find that the air bordering AC cirrus clouds frequently passes the MD activation threshold and reaches up to the CS activation threshold. The probability density over the CS threshold drops significantly with only a very small fraction of the data points reaching values over the HOM threshold. For the WAI cirrus on the other hand, the probability density for supersaturated data points

over the CS activation threshold and also over the HOM threshold is higher. For both cloud types the probability density for supersaturated data points is higher in the near-cloud than the in-cloud area.

The cloud-free air for the two groups is rather similar although there are some differences. Firstly, as also noted in the in-cloud and near-cloud areas, the temperature range of the atmospheric background during AC conditions is narrower, than for the WAI conditions. During AC conditions the cloud-free air is more frequently supersaturated but mostly up to the activation

threshold for MD, whereas during WAI conditions the cloud-free air is less frequently supersaturated, but reaches values over the MD activation threshold and up to the CS threshold.

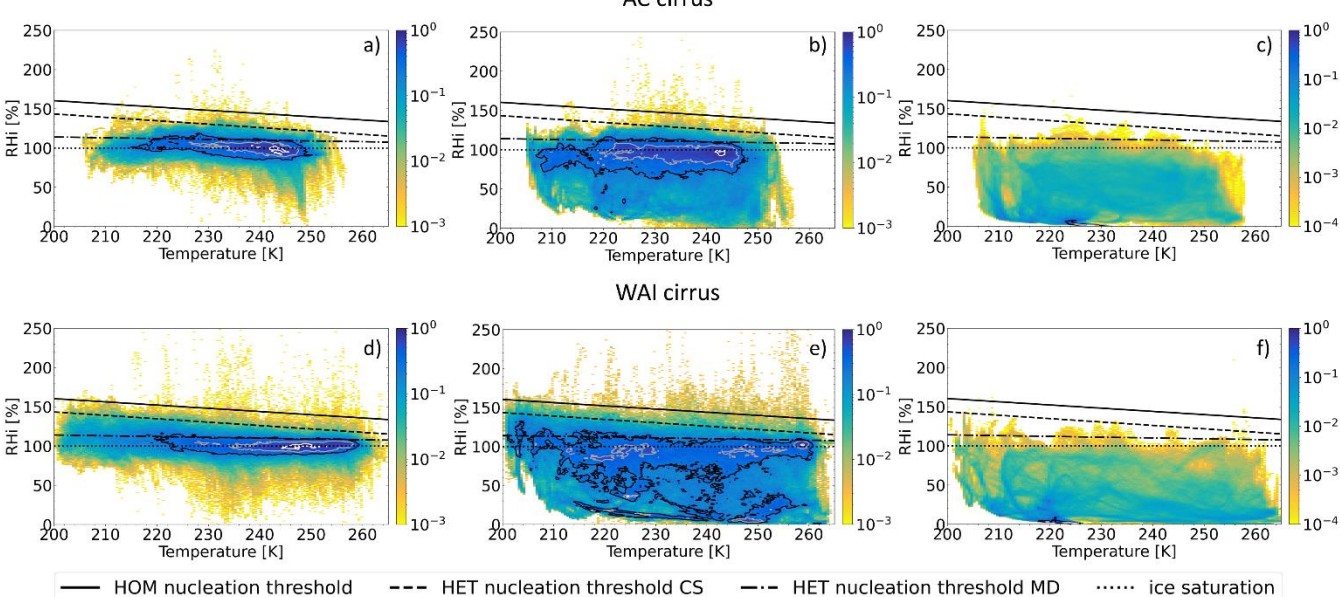

Figure 4: Probability densities of Relative Humidity over ice (RHi) with respect to temperature for: (a) Cirrus clouds measured under arctic conditions (Arctic cirrus), in-cloud (b) Arctic cirrus near-cloud (c) Arctic cirrus cloud-free (d) Cirrus clouds measured under WAI conditions (WAI cirrus), in-cloud (e) WAI cirrus near-cloud (f) WAI cirrus cloud-free. Panels (c) and (f) have a different range in the color scale than the other panels, but same to each other. The bin sizes are 0.5 K and 1 % RHi. The contour lines represent the 0.25, 0.5 and 0.85 probability contours in black, grey and yellow respectively. The dotted line represents the ice saturation threshold (RHi = 100 %). The dash dotted line corresponds to the threshold for heterogeneous nucleation (HET) with mineral dust (MD) as the ice-nucleating particle (INP), which is efficient in initiating ice formation. The dashed line is the threshold for HET nucleation with coated soot (CS) as INP, which is not as easily activated as MD. The solid line denotes the threshold above which homogeneous nucleation (HOM) can take place. Ice crystals can form via HOM nucleation without the need of INPs.





In Table 4, we present a more quantitative temperature independent analysis. We have defined four RHi regimes: subsaturated, low HET, high HET and HOM (See Sect 2.3.4) and calculate the frequency of detection for each bin for the in-cloud, near-cloud and cloud-free regions as defined in Sect. 2.3.3. The HET thresholds here are not connected to specific INPs but rather
generally split HET into two parts with easily and not easily activated INPs.

In Table 4, we show that 57.39% of the data points in WAI cirrus are supersaturated and 53.91 % of AC cirrus. Most of the supersaturated data points for both cloud types are in the HET regime, 57.15 % and 53.87 % for WAI and AC cirrus respectively. The frequencies in the low HET regime are quite similar, 52.12 % and 51.88% for WAI and AC cirrus respectively, but the difference in the high HET regime is bigger, 5.03 % for WAI and 1.99 % for AC cirrus. In the HOM
regime, there is also a higher frequency for the WAI clouds (0.23%) compared to the AC clouds (0.04%).

**Table 4: Probabilities of RHi for the in-cloud, near-cloud and cloud-free regions of the AC and WAI cirrus groups.**

|  | In-cloud | | Near-cloud | | Cloud-free | |
|---|---|---|---|---|---|---|
|  | AC | WAI | AC | WAI | AC | WAI |
| **RHi probability [%]** | | | | | | |
| ≥ 100% | 53.91 | 57.39 | 22.83 | 24.18 | 0.69 | 0.64 |
| **RHi probability [%]** | | | | | | |
| HET (100 % - 147 %) | 53.87 | 57.15 | 22.80 | 23.78 | 0.69 | 0.64 |
|    Low HET (100 % - 123 %) | 51.88 | 52.12 | 22.02 | 19.72 | 0.68 | 0.60 |
|    High HET (123 % - 147 %) | 1.99 | 5.03 | 0.78 | 4.06 | 0.01 | 0.04 |
| HOM ( ≥ 147 %) | 0.04 | 0.23 | 0.04 | 0.41 | 0.00 | 0.00 |

RHi values in the near-cloud area are most frequently subsaturated for both cloud groups. Around WAI cirrus supersaturation is detected with a frequency of 24.18 % while it is 22.83 % around AC cirrus. 23.78% of the WAI cirrus near-cloud data are
in the HET regime and 22.80 % of the AC data. Despite, the near-cloud region of WAI clouds being more frequently in the HET regime overall, the frequency in the low HET regime is slightly higher for the AC cirrus (22.02% against 19.72% for WAI) contrary to the frequency of the high HET regime which is significantly higher for the WAI cirrus (4.06% against 0.78% for AC cirrus). Regarding the HOM regime the frequency for the WAI cirrus is 0.41 % in comparison to 0.04% for AC cirrus. Finally, the cloud-free background conditions are strongly dominated by subsaturated data. Supersaturation is not frequent
during AC or WAI conditions, 0.69 % and 0.64 % respectively. In the low HET regime, the frequency is 0.68%  for the AC conditions and 0.60%  for the WAI conditions respectively. The frequency in the high HET regime is 0.04 % for the WAI conditions and 0.01% for AC conditions. The frequency of detection in the HOM regime is 0 for AC as well as WAI conditions. In summary, we find inside and around AC cirrus very little data points with RHi in the high HET and HOM regimes, consistently lower than for the WAI cirrus with the exception of the near-cloud low HET regime. This could be a possible
indication that these clouds formed via HOM nucleation, thus depleting the higher RHi values. Another possibility would be




that this observation stems from the detection of older stable clouds where new ice formation has seized after either HET or HOM nucleation due to depletion of the available water vapor. On the other hand, the WAI cirrus are detected with more data points in the high HET and HOM regimes compared to the AC cirrus and similar or less points in the low HET regime. This could be an indication either of young newly formed clouds or stable clouds where new ice crystals are still being formed

because of the greater amounts of available water vapor. Both HET and HOM nucleation are possible.

To verify the statistical significance of the results presented in Table 4, a two-sample t-test was conducted. First the variance, skewness and kurtosis of each group where tested to see if such a test could be applied. The t-test returned t-statistics higher than 0 and p-values less than 0.05% for all comparisons.

### 3.2.3 Vertical structure of RHi inside AC and WAI cirrus

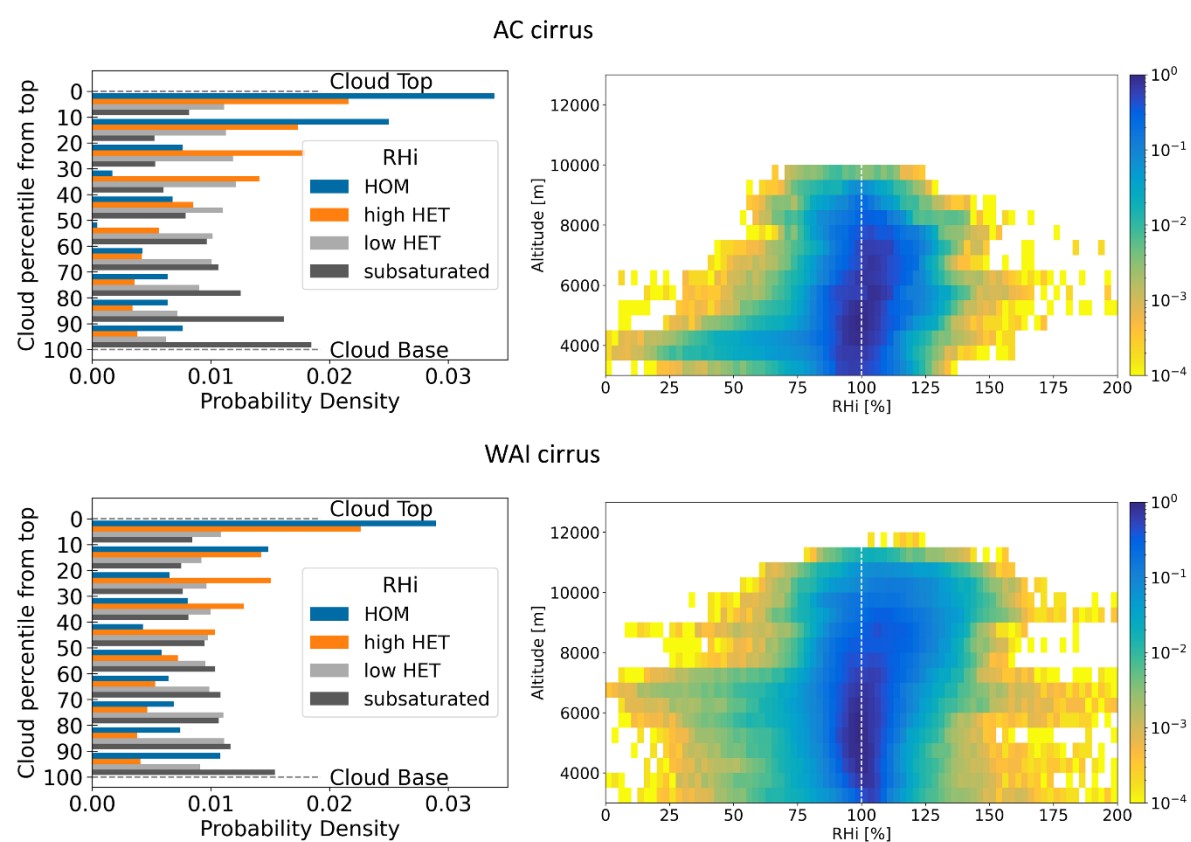

**Figure 5:** Left: Probability densities of in-cloud RHi in a relative location to the cloud top. Every cloud is split into ten percentiles from cloud top to cloud base and for each percentile the probability densities of each RHi bin (subsaturated, low HET, high HET and HOM (See Sect. 2.3.4)) are calculated. Right: Probability densities of in-cloud RHi with respect to altitude. Bin sizes are 2.5% RHi and 500 m altitude. Top row: Cirrus clouds measured under AC conditions. Bottom row: Cirrus clouds measured under WAI conditions

In Figure 5 we present the vertical distribution of RHi within the AC (top) and WAI (bottom) cirrus clouds. We consider the distributions from cloud base to cloud top (left) and with respect to altitude (right). For both groups the cloud tops are dominated by high supersaturations while the cloud bases are mostly populated by subsaturated data points. Although the





general structure is similar a more detailed analysis reveals some differences. AC cirrus are strongly dominated by high supersaturations down to 20 % cloud depth. From 20% to 40% cloud depth the amount of data in the HOM regime reduces

significantly and the clouds have mostly RHi in the high HET regime. The low HET is most dominant from 40% down to 60 % cloud depth. The subsaturated regime starts becoming more and more significant already from cloud middle and becomes dominant from 70% cloud depth down to cloud base. WAI cirrus are also populated mostly by high RHi in the HOM regime in the first 20% of the cloud depth, but already after the first 10% the high HET regime has a significant contribution. The high HET regime is the most populated from 20% cloud depth down to cloud middle. From cloud middle and until 90% cloud depth

the low HET and subsaturated regimes have a similar probability density. The cloud base is clearly dominated by subsaturation although interestingly the HOM regime becomes significant again at this cloud depth.

Regarding the vertical structure with respect to altitude, first we confirm our previous finding, that the WAI cirrus extend higher in the troposphere. AC cirrus seem to be strongly subsaturated for altitudes below 5 km. After that they tend towards high RHi between 5 and 8 km with a peak at around 6km. For WAI cirrus there is a clear trend to high supersaturations towards

higher altitudes. From the cut-off at 3 km and up to 7 km the distribution is rather stable around the ice saturation (RHi =100%) with strong contributions also from subsaturated data. From an altitude of 8 km and up to almost 12 km, supersaturation becomes dominant and higher RHi become more frequent towards higher altitudes.

## 4 Discussion and conclusions

In this study we used water vapor measurements taken during the HALO-(AC)3 campaign by the WALES lidar system and

studied the effects of WAI on the arctic atmosphere and the cirrus clouds that form there. By analyzing the temperature fields from the ERA-5 reanalysis dataset, we found that the high arctic is on average around 8 K warmer during a prevailing WAI event for a layer from the surface up to 500 hPa. At the surface the difference is greater, almost 10 K. Stramler et al., (2011), also define two atmospheric states in the arctic during winter. A cloudy and warmer one, and a clear-sky and cold one. The characteristics of these two states closely correspond to WAI events and an undisturbed arctic as defined in our study. They

find a similar temperature difference at the surface of 13 K. Johansson et al., (2017), used satellite measurements and detected a surface temperature increase by 5.3 K in winter because of airmass intrusion events. At the pressure level of 250 hPa we find the WAI to be slightly colder than the AC, contrary to the other pressure levels. This might be an indication that due to the colder temperatures during AC conditions the troposphere is thinner so that the 250 hPa level is already in the stratosphere, where the temperature tends to increase again.

For the water vapor concentrations, we used both the total column water vapor from ERA-5 as well as our own measurements of water vapor mixing ratio from the WALES lidar system. Both datasets showed on average an increase of up to 2.5 times in the water vapor concentration during prevailing WAI events. From the ERA5 data we found a jump from ~3.4 to ~6.9 kg·m$^{-2}$ and from our own measurements with WALES from ~352 to ~888 ppm. Gierens et al., (2020) find mixing ratios up to 500 ppmv in the ice supersaturated regions of the arctic with the use of radiosondes. Doyle et al., (2011) performed ground-based





measurements for one WAI case and found that the water vapor mixing ratio during the intrusion event was four times higher compared to a background measurement.

The cirrus clouds that form under arctic (AC) and warm air intrusion (WAI) conditions are also found to have differences. Starting with their geometrical characteristics, we found that WAI cirrus are thicker than AC cirrus with a mean depth of around 3141 m and 2037 m respectively. WAI cirrus have a slightly higher cloud base (4990 m compared to 4338m) but extend

on average 1755 m higher with a mean cloud top height of 8131m compared to 6376 m for AC cirrus. This finding could be explained by the warmer conditions during WAI events causing the troposphere to thicken in general allowing clouds to extend to higher altitudes. Devasthale et al., (2011) studied the geometrical properties of cirrus clouds with an optical depth <3 in the arctic or a period of four years. They reported a mean geometrical depth of 0.4 – 1 km with an average cloud base around 6-8 km and an average cloud top around 7-9 km. Nakoudi et al., (2021) used an earthbound Raman lidar at Ny-Ålesund, Svalbard

and radiosondes to measure cirrus properties from 2011 to 2020. They report average cirrus cloud depths of 2.1 km during winter with a cloud base around 7.1 km and a cloud top on average at 9.3 km. Finally, Schäfer et al., (2022) use a ground based and space-born lidar and measure cirrus clouds over the Norwegian arctic for a period of seven years. They find a yearly average cloud thickness of 2.2 km, an average cloud base at 6.9 km and cloud top at 9.1 km. Differences between the reported cloud depths, tops and bases might stem from masking, cloud selections and seasonality as all three publications mention

finding seasonal variations, although not pronounced. Finally, we also calculated the optical depths and we found the WAI cirrus to be optically thicker. The difference in optical thickness can be explained on the one hand because WAI cirrus are also geometrically thicker, but it also hints to differences in the microphysical properties of the two cloud types leading to different extinction coefficients.

Regarding the Relative Humidity over ice (RHi) we found supersaturation both inside and around both AC and WAI cirrus.

From the temperature dependent distribution of RHi inside AC and WAI cirrus, we found WAI cirrus to be consistently over the activation threshold for coated soot, while AC cirrus cross this threshold less frequently. WAI cirrus were also more frequently detected over the threshold for HOM nucleation. The air immediately surrounding WAI and AC cirrus had similar characteristics as the in-cloud. More specifically, the WAI near-cloud data were more frequently detected over the coated soot activation threshold and even the HOM threshold, than the AC cirrus, and for a wider temperature range. The cloud-free air

under AC and WAI conditions was, dominated by subsaturated data points. Interestingly the cloud-free data for the AC group were more frequently supersaturated than the WAI group but remained at lower values of supersaturation, whereas the WAI cloud-free data were less frequently supersaturated but reached RHi values also up to the CS activation threshold.

Furthermore, we defined three regimes, low HET for RHi between 100% and 123%, a high HET regime from 123% to 147 % RHi and a HOM regime for RHi over 147%. We found that WAI clouds were slightly more frequently supersaturated and had

higher frequency of detection both in the HET and the HOM regime, 0.23%. Cziczo et al., (2013) used in-situ measurements over the mid-latitudes and found less than 0.5% of their in-cloud RHi to be over the HOM threshold. In comparison we found 0.04% of the AC and 0.23% of the WAI data points over the HOM threshold. In the near-cloud area supersaturation percentages drop for both AC and WAI cirrus compared to the in-cloud data. The near-cloud area around WAI cirrus has a higher frequency




for both the HET and the HOM regime compared to the AC. Once again there is a bigger difference in the high HET than the
low HET regime. The cloud free air is very strongly subsaturated for both AC and WAI conditions. No HOM data are detected
and AC conditions have slightly higher supersaturation percentage compared to WAI conditions. Under AC conditions there
is a higher frequency for the low HET regime, whereas the high HET regime is more populated for WAI conditions.

To our knowledge this is the first study on the effects of WAI in the arctic on the characteristics of supersaturation in and
around cirrus clouds. Thus, there can be no immediate comparison of our findings. Many previous studies have confirmed
high supersaturations  in and around cirrus clouds although with a focus mostly at mid-latitudes (e.g. Jensen et al., 2001;
Ovarlez et al., 2002; Comstock et al., 2004; Gensch et al., 2008; Krämer et al., 2009; Dekoutsidis et al., 2023). De La Torre
Castro et al., (2023) used in-situ measurements and also found high supersaturations with a median RHi of 125% for cirrus
clouds that were formed and detected in high latitudes, although they set the threshold for clouds to be considered arctic at a
latitude of 60°N. They also conclude that their high latitude clouds formed mainly by HET nucleation in an environment with
few INP and a high RHi, similar to the WAI clouds in our study. Rolf et al., (2022), on the other hand, used in-situ
measurements from the same campaign as De La Torre Castro et al., (2023) and although they also report high supersaturations
inside and outside cirrus clouds they find HOM nucleation as the leading ice-forming process. Finally, Gierens et al., (2020)
analyzed measurements with radiosondes and measured RHi values reaching and sometimes exceeding 150-160% in ice
supersaturated regions.

In summary, we found WAI cirrus and the atmosphere under WAI conditions to be more frequently supersaturated and have
a higher frequency of detection in the high HET and HOM regimes, while AC cirrus and their near-cloud data only have a low
probability of detection in the high HET and HOM regimes with only exception to that the near-cloud low HET regime. It is
expected that homogeneous nucleation produces higher ice number concentrations (Comstock et al., 2008). This in turn leads
to a rapid decrease in RHi due to the deposition of water vapor, thus resulting in lower measured RHi in HOM conditions (e.g.
Jensen et al., 2013, in the tropics). This could lead to the conclusion that AC cirrus formed probably via mainly HOM
nucleation, quickly depleting the available water vapor and resulting in a drop in the RHi. An alternative explanation would
be that older, stable AC cirrus were measured after the bulk of the formation of new ice crystals had already ceased. On the
other hand, the detected WAI cirrus could be considered younger clouds or clouds where new ice crystals are still forming
leading to higher frequencies in the high HET and HOM regimes as they are not yet depleted. Due to the quickly changing
nature of RHi and its reaction to the nucleation processes a definite conclusion cannot be made.

Finally, the vertical structure of RHi within AC and WAI cirrus was studied. In both groups the cloud tops are dominated by
high supersaturations in the HOM regime, while the cloud bases are mostly subsaturated. That being said, there are also some
differences among the two groups. The HOM regime is more dominant in the cloud tops of AC clouds, while lower
supersaturations in the high HET regime start dominating higher in the WAI clouds. Additionally, the cloud bases of AC
clouds were found to be strongly dominated by subsaturation while for WAI clouds there is still a significant contribution by
higher RHi in the HOM regime. While subsaturation becomes very dominant after 70% cloud depth for AC cirrus, the cloud
bases of WAI clouds were found to contain a significant amount of supersaturated data points, indicating potentially a stronger



mixing or sedimentation of ice crystals. Kärcher, (2005) analyze a case study of a polar cirrus which would most probably correspond to an AC cirrus in our study. Similar to our findings, they report a thin highly supersaturated cloud top where ice
crystals form most likely homogeneously. Deeper in the cloud they also detect a gradual decrease in RHi, which stays supersaturated until the cloud base, which is subsaturated. Gierens et al., (2020) also detect ice supersaturations more frequently directly under the tropopause which would coincide in many cases with the cloud tops in our study.

## 5 Summary and outlook

Warm air intrusions (WAI) are events during which warm and moist airmasses from the midlatitudes are transported into the
arctic. In this work we used measurements by the WALES lidar system taken during the HALO-(AC)3 campaign in order to study the effects of warm air intrusions on the arctic atmosphere and the cirrus clouds that form in the high arctic. For this campaign WALES was aboard the German research aircraft HALO. With HALO we were able to reach sufficiently over the cirrus clouds and take measurements deep into the arctic. The WALES lidar system combines both the HSRL and DIAL measurement techniques. It provides collocated measurements of backscatter coefficient, linear depolarization and water vapor
mixing ratio, with high spatial and temporal resolution on a two-dimensional curtain along the flight track. This is a significant advantage over in-situ measurements as we gain information also on the vertical axis, and also over ground-based measurements as we are able to select and follow the desired clouds or systems we want to characterize. Combining the exceptional capabilities of HALO and WALES during the HALO-(AC)3 campaign resulted in the creation of a unique, novel, 2-D dataset of water vapor measurements over the arctic including also data on the vertical axis creating a curtain even of the
whole atmospheric profile under favorable conditions.

From our analysis we found that WAI events do significantly affect the high arctic troposphere and the arctic cirrus clouds. During a WAI event the arctic becomes moister and warmer and the cirrus clouds become geometrically and optically thicker and reach higher altitudes at cloud top. Cirrus clouds measured under WAI conditions are more frequently supersaturated than arctic cirrus (AC) even at the very high supersaturations above the threshold for homogeneous ice nucleation (HOM).
Regarding their vertical structure, AC cirrus clouds have a strongly supersaturated cloud top and a strongly subsaturated cloud base, whereas WAI cirrus clouds show high supersaturations even at cloud base although they tend to higher supersaturations and cloud top.

To our knowledge this is the first study of the water vapor distributions of cirrus clouds in the arctic under WAI conditions. Although lower level clouds have already been studied, cirrus are often overlooked possibly also due to the scarcity of airborne
measurements in the arctic. Understanding the geometrical and macrophysical characteristics of cirrus in the arctic and grasping an estimation of their formation processes is vital in further understanding their microphysics, radiative properties, climate effects and contribution to arctic amplification. Further studying the differences between cirrus clouds in the mid-latitudes and high latitudes is important for a better understanding of arctic amplification and how it is affected by cirrus. The difference in optical depth between WAI and AC cirrus reported in this work is an indication of potential differences also in



the microphysical properties of the two cloud types, which should be further investigated. Combining measurements of macro- and microphysical properties of these clouds along with measurements of the ambient conditions during their formation, would be a key to understand the formation processes of cirrus clouds in the arctic, their life cycles, radiative properties and effects on the climate.

**Data availability**

The data collected by the WALES lidar system during the HALO-(AC)3 campaign that were used in this study can be found at the HALO database: https://doi.org/10.17616/R39Q0T. The ECMWF operational analysis and forecast data were retrieved from the ECMWF Meteorological Archival and Retrieval System (MARS): https://www.ecmwf.int/en/forecasts/access-forecasts/access-archive-datasets (ECMWF, 2022).

**Author contribution**

The study was conceptualized by GD and SG. Data curation was carried out by MW. Formal analysis, investigation, software and visualization were done by GD, assisted by SG and MW. Writing of the original draft was done by GD under the supervision of SG. The review and editing of the manuscript were done by SG and MW.

**Competing interests**

The authors declare that they have no conflict of interest.

**Acknowledgements**

Part of this work was founded by the Deutsche Forschungsgemeinschaft (German Research Foundation, DFG) through the HALO Priority Program SPP 1294, "Atmospheric and Earth System Research with the Research Aircraft HALO (High Altitude and Long Range Research Aircraft)". We also gratefully acknowledge the funding by the DFG - Projektnummer 268020496 - TRR 172, within the Transregional Collaborative Research Center "ArctiC Amplification: Climate Relevant
Atmospheric and SurfaCe Processes, and Feedback Mechanisms (AC)3.
This study contains modified Copernicus Climate Change Service information [2023]. Neither the European Commission nor ECMWF is responsible for any use that may be made of the Copernicus information or data it contains.



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
