# Peer review of "The effects of warm air intrusions in the high arctic on cirrus clouds"

_EGUsphere, 2023_

## Referee Comment (RC2)

The paper presents the effect that warm air intrusions in the high Arctic have on the optical and macrophysical properties of cirrus clouds, and falls within the scope of the journal. The study is based on high-resolution and high-accuracy airborne HSRL data with a focus on water vapor measurements. The study is of high importance as for the first time such a significant amount of water vapor vertical measurements in cirrus clouds are used to investigate this subject, and it is also the first study to study this topic in respect to supersaturation in and around cirrus clouds. The manuscript is well-structured and well-written, clearly presenting the authors' findings. I would suggest the publication of this work after consideration from the authors of the following comments (mostly minor ones).

**Specific comments**
**Page 1 line 15 and Page 3 line 77:** "RHi is affected by and can be used as an indication of the nucleation process and the structure of cirrus clouds" and "the distribution of RHi in the resulting cloud is highly dependent on the nucleation process": Rather than the RHi is being affected by or is dependent on, one would argue that it affects the nucleation processes that will dominate. Suggestion for a (tense) revision.

**Page 2, line 53**: "vertical speed omega": consider revising to updrafts

**Page 5, line 140**: While reading this part I was puzzled why the authors used temperature fields in different levels but wvmr in all atmospheric column (not in levels). Later on, in the results section wvmr in levels is also discussed. So I would suggest here to add (maybe even in parenthesis) the information that later on for the results section wvmr means for different altitude ranges are also discussed/used.

**Figure 2, Page 9 line 202, Table 3**: "Data from ERA-5", "ERA-5 data", and "ERA-5 reanalysis data".. I would suggest that the authors change these parts to ERA-5 reanalysis dataset or ERA-5 fields.

**Page 8, line 140**: "Here, we calculate three temperature-dependent thresholds.. which is more efficient as an INP needing lower RHi": It would be great if the authors could mention here the temperature-dependent thresholds used refer to the RHi values. Also, it would be nice to mention these thresholds so the reader doesn't necessarily search for Urbanek's et al. work before reading the results.

**Page 8, line 181:** "Further, we define three RHi regimes" It seems that these regimes are an output of the previously defined thresholds. If this is the case, please revise the connection between the 2 parts of this paragraph to make it more clear (currently it reads as if this is a new threshold).

**Figure 4 vs section 2.3.5, 3.2.2 (e.g. Page 9 line 230), and Table 4**: It is difficult to understand the connection between the thresholds presented in the plots/legends and the ones mentioned in section 2.3.5 and page 9 line 230. Specifically, in the text 3 regimes are described, the HOM nucleation regime defined with values over 147%, HET CS between 123% and 147%, and HET MD between 100% and 123%. On the contrary in Figure 4 the thresholds are discussed differently, than the ones mentioned in the main text, with the lower values of HET MD being higher than 100% and this propagates somehow in the other thresholds. Please revise to make clear and consistent the different regions in fig. and main text.

**Page 11, line 259:** "During AC conditions the cloud-free air is more frequently supersaturated but mostly up to the activation threshold for MD, whereas during WAI conditions the cloud-free air is less frequently supersaturated, but reaches values over the MD activation threshold and up to the CS threshold": This is not visible when comparing the © and (f) plots by eye. I

assume this is discussed base on the statistics of table 4, so maybe you could add the a reference on the table for this part.

Page 15 line 346: "The difference in optical thickness … hints to differences in the microphysical properties of the two cloud types leading to different extinction coefficients": Did you find different extinction coefficients from the collocated WALES extinction measurements? if so, a comment on this would be interesting to be included.

**Technical corrections/suggestions (bold text & red ","):**
Page 3, line 85: "..high ice supersaturation **values in their simulations**".
Page 4, line 113: "the measurements come in **the** form of… the whole atmospheric column**,** from the aircraft to the earth's surface**,** when conditions are favorable. Thus, with a single overpass of a cloud**,** information ..".
Page 5, line 138: "For the temperature field we used  the pressure level".
Page 8, line 179: "and thus  higher RHi values **are** necessary".
Page 8, line 196: "until  the 250 hPa pressure level **where** the arctic becomes"
Page 8, line 140: "Here, we calculate three temperature-dependent **RHi** thresholds"
Page 9, line 225: "WAI cirrus have  almost".
Figure 4: "under arctic conditions ( AC), in-cloud (b)  AC near-cloud (c)  AC cloud-free (d) Cirrus clouds measured under WAI conditions (WAI ), in-cloud (e) WAI  near-cloud (f) WAI  cloud-free." Because it is confusing reading "cirrus cloud-free" in the legend.

---

## Author Comment (AC1)

**Referee #1**

Dear Referee 1,

speaking for all co-authors I would like to thank you for putting your time and effort into this review. Thank you also for your very positive comments. In the following I will try to address all your comments and questions in detail. Your comments/questions are presented in Bolt lettering followed each by the reply. For changes in the text I present the original text in blue and the changed or added text in green color.

**General comments**

1. **Given that the paper is about differences between undisturbed arctic conditions (AC) and prevailing warm air intrusion conditions (WAI), I was expecting more details about flights grouping. "We studied the synoptic situation for each flight", as written in Sect. 2.3.1, should be developed. What are the criteria for the classification? Please explain the information provided in Fig. 2.**

   More details have been added to Sect 2.3.1 and it has been changed as follows:

   In this study we aim to detect differences in cirrus clouds measured in the high arctic during undisturbed arctic conditions (AC) and during prevailing warm air intrusion conditions (WAI). In order to accurately identify the ambient conditions as AC and WAI and correctly group each flight into one of the two categories, various sources and parameters were used. The first step was already taken during the campaign. Daily weather briefings were held for the flight planning. Discussed during these briefings were amongst others the surface analysis, the development of circulation patterns, the wind fields and airmass advections as well as more specific forecasting indexes like the vertically integrated water vapor transport (IVT). The IVT is calculated from the surface and up to 300 hPa and is used to define WAIs and atmospheric rivers with a common threshold being 250 kg·m$^{-1}$·s$^{-1}$. This already allowed an early classification of the flights. Additionally, after the campaign we once again studied the synoptic situation for each flight, using the geopotential height, temperature and total column water vapor fields from the ERA5 reanalysis dataset (Hersbach et al., 2018), as well as the water vapor measurements from WALES. Flights during which the general circulation would allow for the transport of airmasses from the mid-latitudes and elevated temperatures and water vapor concentrations were detected, were classified as WAI. Despite it not being set as a threshold during the selection, all AC flights had a mean water vapor mixing ratio strongly below 500 ppm as measured by WALES, while WAI flights had a mean strongly above 500 ppm. In Fig. 2 an example of the performed analysis is presented showcasing on the right side the geopotential heights allowing the northward transport and elevated water vapor concentrations. Henceforth, we shall refer to the cirrus clouds of the AC and WAI groups as arctic cirrus and WAI cirrus respectively. In order to be more precise in our classification parts of some flights where discarded, mostly due to HALO exiting the WAI events when performing turns.

2. **In Table 3, what does "± value" mean? Is "value" a standard deviation or other?**

   The ± value is indeed the standard deviation. Clarification has been added to the description of the table:

Table 3: Differences in available water vapor, temperature and cloud geometric and optical characteristics during research flights of the HALO-(AC)3 campaign under arctic (AC) and warm air intrusion (WAI) conditions. Overview from ERA-5 reanalysis data and data from WALES measurements. Presented in value ± standard deviation.

3. **Figure 3 is a very good illustration of the cloud masking. I note that this illustration is for flight RF17, which is in the AC category according to Fig. 2. I see cloud tops up to 9.5 km, which seems in contradiction with the fact that AC clouds have top height = 6376 ± 1195 m in Table 3. Please clarify. Which part of the flight do we see in Fig. 3 in terms of latitude and longitude ranges?**

For every time step during the flight, the in-cloud mask is used to define the cloud top and cloud base. Each flight is then assigned the average value and standard deviation. The value in Table 3 is the mean of all flights and the respective pooled standard deviation which could explain this well-noted discrepancy.

Fig. 3 contains all the available data from the flight as it is shown in Fig. 2. Fig. 3 has been renewed and now includes the information on latitude and longitude. A clarification has also been added to the caption of Fig. 3:

[Figure]

Figure 3: Example of cloud mask application. (a) Complete RHi curtain with cloud mask outline (b) In-cloud data (c) Near-cloud data and (d) cloud-free data. The black lines correspond to the mask parameters. The color scale represents RHi. White areas have no data. HALO-(AC)3 campaign research flight 17 on 11/04/2022, including the whole flight as shown in Fig. 2.

4. **Regarding Table 3 and related text, it would be informative to show ECMWF temperature vs. altitude in AC and WAI conditions (perhaps add a figure?). I suggest comparing temperatures at base and top heights.**

Plots were created for the two groups representing the probability densities of temperature with respect to altitude from ECMWF IFS data. The figure will be added as a supplement.

[Figure]

Supplement Figure 1: Probability density function of ambient temperature with respect to altitude for AC Cirrus group (Left) and WAI Cirrus group (Right). Also plotted are the respective average cloud base and cloud top heights for each group. Noticeable are the higher altitude of the thermal tropopause under WAI conditions and the generally warmer conditions especially at lower altitudes. Temperature data from ECMWF IFS.

5. **Nucleation process:**
   o **Unlike in Dekoutsidis et al. (2023), which present a similar analysis but by comparing in situ and liquid origin cirrus clouds at mid-latitude, cloud temperature is not required to be smaller than 235 K (according for instance to Table 2 and as seen in Fig. 4), and it seems that the statistics provided in Table 4 and the discussions are by considering all temperatures. The wording "HOM regime" is misleading because it seems to actually mean "RHi > 147 %", and the line for HOM threshold extending up to 265 K in Fig. 5 is confusing. Please clarify.**

   For the analysis of cirrus clouds over the midlatitudes the threshold at 235 K was seen as necessary for the cloud mask in order to filter out non-cirrus clouds at higher altitudes. For the HALO-(AC)3 measurements this was not necessary as an altitude threshold at 3km was found to be sufficient in filtering out mixed-phase and liquid clouds closer to the surface. An additional threshold at 235 K would in many cases remove warmer parts of cirrus clouds were HET nucleation is still possible.

   The temperature dependent analysis of RHi based on Figure 4 and the nucleation thresholds presented therein are not immediately connected to the analysis in the four regimes.

   Regarding Figure 4, the water saturation threshold has been added to each panel thus more strictly denoting the temperature and RHi ranges for the different nucleation processes. Namely, the 235 K for HOM nucleation. A small fraction of the data in the in-cloud and near-cloud areas is detected over the water saturation line as was the case also in Ovarlez et al., (2002) who performed in-situ measurements. Similar to them we also conclude that these are possibly artefacts or erroneous data, but represent a too small amount of the whole data set to have an effect on the further

[revised manuscript text omitted]

Regime names have been changed on every instance.

o  **In Figure 5 and related text, as well as in Sects. 4 and 5, the interpretation in terms of "HOM regime" is complicated by the fact that temperatures might be larger than 235 K. It is likely that T < 235 K near cloud top, but it is no sure near cloud base. Additional information related to temperature seems necessary.**

Same also to the previous comment we have changed the naming of our RHi regimes so that it is clear that it only refers to RHi values and not specific nucleation processes as such an assumption could be erroneous. Figure 5 has been changed accordingly, and is shown in the following, as has been the text referring to it. On the topic of the temperature a figure has been added to the supplementary material of this publication as also discussed in comment Nr. 4.

[Figure]

Figure 5: Left: Probability densities of in-cloud RHi in a relative location to the cloud top. Every cloud is split into ten percentiles from cloud top to cloud base and for each percentile the probability densities of each RHi bin (subsaturated, low RHi, mid RHi and high RHi (See Sect. 2.3.4)) are calculated. Right: Probability densities of in-cloud RHi with respect to altitude. Bin sizes are 2.5% RHi and 500 m altitude. Top row: Cirrus clouds measured under AC conditions. Bottom row: Cirrus clouds measured under WAI conditions.

    o   **Can the authors clarify the statement in lines 397-398?**

With this statement we attempt to provide a possible explanation regarding the high RHi values detected near cloud base mostly for the WAI cirrus could which would be an increase in the available water vapor. From the previous analysis it is know that during WAI events a significant amount of water vapor is transported into the arctic as well as warm airmasses which tend to rise in some cases with high updrafts. This combined with sedimentation of ice crystals into this region would support the mixing of the airmasses, providing the extra water vapor from lower altitudes or result in sublimation of the ice crystals as another way of increasing the water vapor and thus the RHi.

6.   **Sections 4 and 5 could be shortened, I felt that there were repetitions.**

Sections 4 and 5 have been shortened by removing repetitions.

**Other comments**

1. **Line 11: backscatter coefficient: at which wavelength? 532 nm?** The backscatter coefficient is determined from the signal at 532 nm. A clarification has been added at line 110 (possible typo).
2. **Line 159: please define the backscatter ratio** Definition has been added to the text.
3. **Line 163: PLDR > 20 % => could mixed-phase clouds be included at the warmest temperatures?** There is always this possibility especially for the WAI cases, that is why the altitude has also been included as a parameter of the cloud mask. After analyzing the lidar and radar measurements we are confident that no mixed phase clouds have been included in the cloud mask.
4. **Lines 220-222: can you include a reference for this study? And/or for HAMP?** This short study has been performed as an auxiliary to this publication and is not published in its self. A reference for the HAMP radar has been added.
5. **Lines 232, line 263, and caption of Fig. 5: should Sect. 2.3.4 be 2.3.5?** Yes, has been corrected.
6. **Lines 264: should Sect. 2.3.3 be 2.3.4?** Yes, has been corrected.
7. **Lines 371-375: is there evidence that high latitude cirrus clouds in De La Torre Castro et al. (2023) are WAI? Note that in this study, T is smaller than 235 K.** De La Torre Castro et al. (2023) do not include any information regarding the presence of a WAI, but comparing their findings to ours we find similarities between their arctic cirrus and our WAI cirrus measured in the arctic.
8. **Line 375: do you confirm that the "Rolf et al. (2022)" reference is a short abstract?** Yes, that is confirmed.

**Technical comments**

1. **Fig.1: typo: collumn => column**         Corrected.
2. **Fig. 5: top left panel: we cannot see the value for high HET (orange) in the 20-30 % range. Move the caption?**

   Caption has been moved to better spot.

3. **Lines 359-360: the sentence ends with ",0.23%". Can you rephrase?**

   This was a typo and has been removed.

4. **Kärcher, B.: Supersaturation, dehydration, and denitrification in Arctic cirrus, Atmos. Chem. Phys., 2005 => the reference is incomplete**

   Reference has been corrected.

---

## Author Comment (AC2)

**Referee #2**

Dear Dr. Marinou,

on behalf of all co-authors I would like to thank you for reviewing our submitted manuscript and providing constructive feedback. Thank you also for your very positive opening comments.

In the following please find a detailed reply on each of your comments. For ease of understanding, in cases the text of the manuscript was modified, I present the original text in blue and the updated/added text in green color.

**Specific comments**

- **Page 1 line 15 and Page 3 line 77:** "RHi is affected by and can be used as an indication of the nucleation process and the structure of cirrus clouds" and "the distribution of RHi in the resulting cloud is highly dependent on the nucleation process": Rather than the RHi is being affected by or is dependent on, one would argue that it affects the nucleation processes that will dominate. Suggestion for a (tense) revision.

  Tenses have been revised as follows:

  Ice formation occurs at certain RHi values depending on the dominant nucleation process taking place. RHi can thus be used as an indication of the nucleation process and the structure of cirrus clouds.

  Nevertheless, the nucleation process taking place is strongly correlated to the distribution of RHi in the resulting cloud, as for example HOM needs a much higher RHi to initiate

- **Page 2, line 53:** "vertical speed omega": consider revising to updrafts

  Has been revised as suggested.

  The nucleation process of each cloud is largely dictated by the ambient conditions, e.g. updraft, available INPs, available moisture and ambient temperature

- **Page 5, line 140:** While reading this part I was puzzled why the authors used temperature fields in different levels but wvmr in all atmospheric column (not in levels). Later on, in the results section wvmr in levels is also discussed. So I would suggest here to add (maybe even in parenthesis) the information that later on for the results section wvmr means for different altitude ranges are also discussed/used.

  Following phrase added to this section:

  … flight track of each flight. Results regarding the water vapor mixing ratio are also discussed on four levels from 0 km to 12 km.

- **Figure 2, Page 9 line 202, Table 3:** "Data from ERA-5", "ERA-5 data", and "ERA-5 reanalysis data".. I would suggest that the authors change these parts to ERA-5 reanalysis dataset or ERA-5 fields.

Proposed changes adopted for all suggested points and similar occasions throughout the text.

- **Page 8, line 140**: "Here, we calculate three temperature-dependent thresholds.. which is more efficient as an INP needing lower RHi": It would be great if the authors could mention here the temperature-dependent thresholds used refer to the RHi values. Also, it would be nice to mention these thresholds so the reader doesn't necessarily search for Urbanek's et al. work before reading the results.

Following table including the thresholds added to the manuscript:

Table 1: Temperature dependent thresholds for HOM nucleation and HET nucleation on mineral dust (MD) and coated soot (CS). Temperatures in K

| | |
|---|---|
| $RH_i^{HOM} = 232\,\% - 0.37\,\% \times T\ K^{-1}$ | Parametrization from (Urbanek et al., 2017) based on (Koop et al., 2000) |
| $RH_i^{HET\ (MD)} = 134\,\% - 0.1\,\% \times T\ K^{-1}$ | From (Krämer et al., 2016) |
| $RH_i^{HET\ (CS)} = 230\,\% - 0.43\,\% \times T\ K^{-1}$ | From (Krämer et al., 2016) |

- **Page 8, line 181:** "Further, we define three RHi regimes" It seems that these regimes are an output of the previously defined thresholds. If this is the case, please revise the connection between the 2 parts of this paragraph to make it more clear (currently it reads as if this is a new threshold).

This was also identified as a problem by referee 1 and has been adequately addressed in the manuscript.

- **Figure 4 vs section 2.3.5, 3.2.2 (e.g. Page 9 line 230), and Table 4**: It is difficult to understand the connection between the thresholds presented in the plots/legends and the ones mentioned in section 2.3.5 and page 9 line 230. Specifically, in the text 3 regimes are described, the HOM nucleation regime defined with values over 147%, HET CS between 123% and 147%, and HET MD between 100% and 123%. On the contrary in Figure 4 the thresholds are discussed differently, than the ones mentioned in the main text, with the lower values of HET MD being higher than 100% and this propagates somehow in the other thresholds. Please revise to make clear and consistent the different regions in fig. and main text.

This has also been brought to our attention by referee 1 and changes have been made to address the problem.

- **Page 11, line 259:** "During AC conditions the cloud-free air is more frequently supersaturated but mostly up to the activation threshold for MD, whereas during WAI conditions the cloudfree air is less frequently supersaturated, but reaches values over the MD activation threshold and up to the CS threshold": This is not visible when comparing the © and (f) plots by eye. I assume this is discussed base on the statistics of table 4, so maybe you could add the a reference on the table for this part.

The following phrase has been added to the end of this paragraph:

CS threshold. The differences are made clearer in the further statistical analysis.

- **Page 15 line 346**: "The difference in optical thickness … hints to differences in the microphysical properties of the two cloud types leading to different extinction coefficients": Did you find different extinction coefficients from the collocated WALES extinction measurements? if so, a comment on this would be interesting to be included.

The extinction coefficients on their own are not analyzed in this study. Explanatory text has been added to Sect 2.2 and 4 clarifying that the optical thickness is calculated from the two-way optical transmission due to particle extinction.

**Technical corrections/suggestions (bold text & red ","):**

- Page 3, line 85: "..high ice supersaturations **values in their simulations**".
- Page 4, line 113: "the measurements come in **the** form of… the whole atmospheric column**,** from the aircraft to the earth's surface**,** when conditions are favorable. Thus, with a single overpass of a cloud**,** information ..".
- Page 5, line 138: "For the temperature field we used  the pressure level".
- Page 8, line 179: "and thus  higher RHi values **are** necessary".
- Page 8, line 196: "until  the 250 hPa pressure level **where** the arctic becomes"
- Page 8, line 140: "Here, we calculate three temperature-dependent **RHi** thresholds"
- Page 9, line 225: "WAI cirrus have  almost".
- Figure 4: "under arctic conditions ( AC), in-cloud (b)  AC near-cloud (c)  AC cloud-free (d) Cirrus clouds measured under WAI conditions (WAI ), in-cloud (e) WAI  near-cloud (f) WAI  cloud-free." Because it is confusing reading "cirrus cloud-free" in the legend.

All technical corrections have been applied in the manuscript